# MiRNA-Regulated Pathways for Hypertrophic Cardiomyopathy: Network-Based Approach to Insight into Pathogenesis

**DOI:** 10.3390/genes12122016

**Published:** 2021-12-18

**Authors:** German Osmak, Natalia Baulina, Ivan Kiselev, Olga Favorova

**Affiliations:** 1Laboratory of Functional Genomics of Cardiovascular Disorders, National Medical Research Center for Cardiology, 121552 Moscow, Russia; tasha.baulina@gmail.com (N.B.); kiselev.ivan.1991@gmail.com (I.K.); olga.favorova@gmail.com (O.F.); 2Laboratory of Medical Genomics, Pirogov Russian National Research Medical University, 117997 Moscow, Russia

**Keywords:** hypertrophic cardiomyopathy, human, myocardium, miRNA (microRNA), network analysis, pathway analysis, TGFβ-mediated SMAD signaling

## Abstract

Hypertrophic cardiomyopathy (HCM) is the most common hereditary heart disease. The wide spread of high-throughput sequencing casts doubt on its monogenic nature, suggesting the presence of mechanisms of HCM development independent from mutations in sarcomeric genes. From this point of view, HCM may arise from the interactions of several HCM-associated genes, and from disturbance of regulation of their expression. We developed a bioinformatic workflow to study the involvement of signaling pathways in HCM development through analyzing data on human heart-specific gene expression, miRNA-target gene interactions, and protein–protein interactions, available in open databases. Genes regulated by a pool of miRNAs contributing to human cardiac hypertrophy, namely hsa-miR-1-3p, hsa-miR-19b-3p, hsa-miR-21-5p, hsa-miR-29a-3p, hsa-miR-93-5p, hsa-miR-133a-3p, hsa-miR-155-5p, hsa-miR-199a-3p, hsa-miR-221-3p, hsa-miR-222-3p, hsa-miR-451a, and hsa-miR-497-5p, were considered. As a result, we pinpointed a module of TGFβ-mediated SMAD signaling pathways, enriched by targets of the selected miRNAs, that may contribute to the cardiac remodeling in HCM. We suggest that the developed network-based approach could be useful in providing a more accurate glimpse on pathological processes in the disease pathogenesis.

## 1. Introduction

Hypertrophic cardiomyopathy (HCM) is the most common hereditary heart disease, with an estimated prevalence of 1:500–1:200 [1]. Despite the established role of mutations in sarcomere protein genes in the development of HCM, targeted sequencing of these genes makes it possible to acquire a molecular diagnosis in less than 50% of cases [2], and the rapid spread of high-throughput sequencing has led to a re-estimation of the pathogenicity of known genetic variants towards its reduction and has increased the number of patients with HCM with an unknown cause [3]. In addition, modern transcriptomic studies cast doubts on the monogenic nature of the disease, which suggests the presence of sarcomere-independent non-genetic mechanisms of HCM development [2,4,5]. From this point of view, HCM may arise from the interactions of several HCM-associated genes, and from a disturbance in the regulation of their expression.

MicroRNAs (miRNAs) are small non-coding regulatory RNA molecules; in most cases they downregulate gene expression via incomplete or complete complementary binding with their target mRNAs, causing the repression of mRNA translation or degradation, respectively. As a result, miRNAs are implicated in the fine regulation of a diverse spectrum of physiological processes; miRNAs’ dysregulation is associated with many pathological processes in humans, including hypertrophy, fibrosis, and apoptosis of myocardial cells [6,7]. miRNAs have been shown to be involved in the development of various cardiovascular diseases, such as atherosclerosis, myocardial ischemia, and heart failure (reviewed in [8,9,10,11,12]).

It is known that miRNA target genes form interaction networks; some of these genes (hubs and/or bottlenecks) are nodes of high centrality and they provide rapid coordination between different parts of the miRNA-target network [13]. It is highly likely that the expression of these genes will most sensitively reflect changes in the network. Hereinafter, such genes will be referred to as key target genes, or more simply as key genes. Previously, we proposed an approach that allows highlighting the key genes from miRNA-target networks and subsequent identification of the signaling pathways enriched by these genes [14].

In the present study we used the same algorithm to search for HCM-associated signaling pathways regulated not by a single miRNA but rather by a pool of miRNAs. Considering miRNAs in the context of their cooperative and redundant effects on target genes [15], we hypothesized that several miRNAs may regulate the same segments of signaling pathway networks, uncovering their fundamental importance in disease-driven processes. Thus, to investigate miRNA-mediated signaling pathways in the context of HCM we selected a set of miRNAs based on the data on their contribution to cardiac hypertrophy. We considered only those targets of selected miRNAs that are expressed in human heart tissue to provide a more accurate glimpse of the pathological processes in the heart muscle involved in HCM. By using the developed workflow, we identified shared signaling pathways regulated by studied miRNAs that more likely contribute to HCM pathogenesis.

## 2. Methods

### 2.1. Workflow

The workflow of the study is shown in Figure 1. As the first step, we constructed a separate interaction network for target genes of each studied miRNA, which are expressed in heart tissue. The protein products of these genes served as nodes. An edge was drawn between any pair of nodes if the “minimum required interaction score” by STRING database (https://string-db.org/, accessed on 18 December 2020) was not less than 0.9 with default settings (“highest confidence” in STRING database terminology). The largest connected components (LCCs) were extracted from the constructed networks. For each node in the LCC, the normalized degree and betweenness centrality were calculated. Further, we removed the nodes from each LCC one-by-one in order from the highest sum of both centralities to the lowest. By removing the nodes, the LCC gradually disintegrated into separate connected components, and its cardinality (the number of nodes included in the LCC) dropped. At the moment when the active decay of LCC stopped and LCC cardinality reached a plateau, we stopped the process and considered all the removed nodes as key genes essential for the connectivity of the constructed network. More detailed description can be found in [14].

For the identified key genes, we performed overrepresentation analysis in Reactome pathways using the Reactome2py v.1.0.1 package for Python [16]; the algorithm and program code are described in [14].

Using two sets, “miRNAs” and “Reactome pathways”, linked through key genes, we obtain an injective function: “miRNAs key genes → Reactome pathways”. Taking sets “miRNAs” and “Reactome pathways” as nodes of the network, and the set of “miRNA key genes → Reactome pathways” mappings as edges, we build a bigraph, or miRNA-Pathway network. If the constructed network contains at least two connected miRNAs, i.e., miRNAs connected to each other through the sequence of edges and other nodes, we search for the minimum number of nodes from the set “Reactome pathways” linked to the largest number of miRNAs using the greedy algorithm. If the constructed network does not contain connected miRNAs, we construct a new bigraph with the set of the key genes instead of the set “Reactome pathways”, carry out the search for the key genes shared by studied miRNAs using the same greedy algorithm, and analyze functions of these genes. If the key genes shared by studied miRNAs are absent, we analyze the functions of each miRNA separately, as described in [14].

### 2.2. Tools and Databases

We considered only the data on heart-specific RNA expression, miRNA-target genes interactions, and protein–protein interactions available for *Homo sapiens*.

Networks were constructed using the NetworkX 2.0 package (Pasadena, CA, USA) for Python [17], and their characteristics were visualized using Cytoscape software (Seattle, VA, USA) [18]. The miRTarBase database (accessed on 18 December 2020 [19]) was used to select experimentally validated direct target genes of miRNAs, described for humans. The Human Protein Atlas database (https://www.proteinatlas.org/, accessed on 20 March 2020 [20]) was used to extract data on the expression of protein-coding genes in heart muscle. The STRING database (https://string-db.org/, accessed on 18 December 2020 [21]) was used to search for molecular interactions between protein products of miRNA target genes; only interactions with a high confidence score (score > 0.9) were selected for further analysis.

When performing statistical tests, *p*-values < 0.05 were considered statistically significant.

### 2.3. Availability of Data and Materials

The code is available at GitHub: https://github.com/GJOsmak/miRNET_HCM (accessed on 13 December 2021).

## 3. Results

### 3.1. Identification of Key Target Genes and Analysis of Their Overrepresentation in Reactome Gene Sets

In order to select miRNAs for the analysis we performed a search of studies, which were indexed by the terms “cardiac hypertrophy”, “myocardium”, and “miRNA expression”, in the PubMed database, published by December 2021. Only miRNAs shown to be deregulated in hypertrophic myocardium in two or more studies were considered. As a result, 12 miRNAs, namely hsa-miR-1-3p, hsa-miR-19b-3p, hsa-miR-21-5p, hsa-miR-29a-3p, hsa-miR-93-5p, hsa-miR-133a-3p, hsa-miR-155-5p, hsa-miR-199a-3p, hsa-miR-221-3p, hsa-miR-222-3p, hsa-miR-451a, and hsa-miR-497-5p, were selected for the subsequent analysis (Appendix A).

According to the workflow, for all selected miRNAs, namely hsa-miR-29a, hsa-miR-93, hsa-miR-133a, hsa-miR-199-3p, hsa-miR-221, hsa-miR-222, hsa-miR-451, and hsa-miR-497, interaction networks of target genes were constructed and key genes were extracted from these networks (edge tables for their LCCs, and dictionary {miRNA: key genes} as JSON file are presented in Appendix A, respectively).

Data on the number of miRNA target genes are shown in Table 1. The greatest number of targets is described for hsa-miR-93, the lowest number is known for hsa-miR-451. It is noteworthy that the percentage of targets for all studied miRNAs expressed in human heart tissue ranges from 50 to 60; due to this, the sizes of constructed networks are significantly reduced and their analysis is facilitated. For hsa-miR-133a, hsa-miR-199a-3p and hsa-miR-451 more than half of their targets, included in LCCs of the networks, are identified as key genes of LCCs. For the rest of the miRNAs, the number of the key genes is significantly less than the size of their LCCs. It can be noted that the greatest number of targets (16%) were identified as key genes for hsa-miR-451, regulating the smallest number of genes; for the rest of the miRNAs, the algorithm proposed in [11] allowed us to narrow down the number of analyzed targets (key genes) to less than or equal to 6% of its total number.

Reactome overrepresentation analysis, carried out separately for key genes of each miRNA, showed that hsa-miR-1-3p targets are overrepresented in 25 Reactome gene sets, hsa-miR-19b-3p—in 50, hsa-miR-21-5p—in 43, hsa-miR-29a-3p—in 38, hsa-miR-93-5p—in 64, hsa-miR-133a-3p—in 12, hsa-miR-155-5p—in 60, hsa-miR-199a-3p—in 59, hsa-miR-221-3p—in 116, hsa-miR-222-3p—in 39, hsa-miR-451a–in 49 and hsa-miR-497-5p–in 46 gene sets. Dependency tables for miRNAs and corresponding Reactome gene sets are presented in Appendix A.

### 3.2. Greedy Search for Pathways Shared by Studied miRNAs

Following the workflow, we constructed the miRNA-pathway network for studied miRNAs and obtained a connected network in which all miRNAs were present (upper network on Figure 2A).

We assumed that if selected miRNAs have a shared regulatory effect on signaling pathways, then the observed degree distribution in the connected network should be skewed to the right relative to the expected distribution for a randomly selected similar-sized set of miRNAs. To test this statistical hypothesis, we estimated the probability of obtaining similar results with twelve random miRNAs using the Monte Carlo method (10,000 iterations). The analysis showed the shift of the observed degree distribution relative to the expected: the observed number of nodes with high degree centrality reflecting signaling pathways regulated by more miRNAs is estimated to be higher than what was expected, and vice versa for pathways regulated by fewer miRNAs (*p*-value = 5.9 × 10^−4^, for calculation see Appendix A). Therefore, we affirmed that the miRNA-pathway interaction network was not obtained by chance.

We used the greedy algorithm to search for Reactome pathways connected to the majority of the selected miRNAs (Figure 2A). At the first iteration of the algorithm, we pinpointed a module shown in Figure 2B that contains three pathways involved in TGFβ-mediated SMAD signaling (R-HSA-2173796, R-HSA-2173793, and R-HSA-170834) and overrepresented by key target genes of eight miRNAs, namely hsa-miR-19b-3p, hsa-miR-21-5p, hsa-miR-29a-3p, hsa-miR-93-5p, hsa-miR-155-5p, hsa-miR-221-3p, hsa-miR-222-3p, and hsa-miR-451a. For a random set of twelve miRNAs the probability of obtaining the module containing not less than eight miRNAs targeting these pathways is 0.0062 (for calculations see Appendix A). Notably, when we performed a preliminary run of greedy search using a reduced list of miRNAs (hsa-miR-29a, hsa-miR-93, hsa-miR-133a, hsa-miR-199-3p, hsa-miR-221, hsa-miR-222, hsa-miR-451, and hsa-miR-497), and not taking into account their -3p/-5p strand specificity, greedy search similarly pinpointed TGFβ-mediated SMAD signaling at its first iteration (Appendix A).

To perform the second iteration of the greedy search we excluded the TGFβ-mediated SMAD signaling module and connected miRNAs from bigraph. As a result, we pinpointed another module consisting of 12 pathways; each of them was overrepresented by key targets of two miRNAs (Figure 2C). The first cluster in the module, consisting of pathways 1–4, is regulated by hsa-miR-1-3p and hsa-miR-497-5p and involved in transport of mature transcripts to cytoplasm (R-HSA-72202), histone modification (R-HSA-3214858), and meiosis (R-HSA-1500620 and R-HSA-912446). The second cluster of pathways 5–10 is regulated by hsa-miR-133a-3p and hsa-miR-199a-3p and involved in VEGF signaling (R-HSA-4420097 and R-HSA-194138), transcriptional regulation by TFAP2 (R-HSA-8864260 and R-HSA-8866910), and cellular response to hypoxia (R-HSA-1234158 and R-HSA-1234174). The remaining two “bottleneck” pathways of oxidative stress-induced senescence (R-HSA-2559580) and activation of PI3K/AKT signaling (R-HSA-8851907) are regulated by hsa-miR-199a-3p together with hsa-miR-497-5p or hsa-miR-1-3p. However, the probability of obtaining such a module by chance is equal to 0.11 (for calculation see Appendix A); therefore, only the first module of TGFβ-mediated SMAD signaling pathway will be further considered.

### 3.3. Analysis of miRNA Interactions with Key Targets Included in TGFβ-Mediated SMAD Signaling Pathways

For a more precise understanding of the regulatory role of the miRNAs from the module containing pathways that are involved in TGFβ-mediated SMAD signaling, we analyzed interactions between these miRNAs and their key targets (Figure 3). The interaction network shows that MYC is the most targeted gene; its expression is regulated by hsa-miR-21-5p, hsa-miR-29a-3p, hsa-miR-93-5p, hsa-miR-155-5p, hsa-miR-222-3p, and hsa-miR-451a. Genes *MAPK1* and *SMAD4* are regulated by three miRNAs (hsa-miR-19b-3p, miR-935p, and miR-451a or miR-155-5p, respectively). Genes *RNF111*, *CCNT2*, *SP1*, *RHOA*, and *UBC* are targeted by sets of two different miRNAs. Each of the rest of the genes, namely *XPO1*, *SMAD3*, *SMAD7*, *STUB1*, and *RPS27A*, is regulated by a single miRNA. Hsa-miR-155-5p, hsa-miR-93-5p, hsa-miR-21-5p, and hsa-miR-19b-3p, each of which interacts with at least four genes of the constructed bigraph, have the most regulatory effect on TGFβ-mediated SMAD signaling pathway. Other miRNAs target 1–2 key genes in the network.

The results we obtained, when we analyzed miRNA interactions with key targets within the second module, are presented in Appendix A.

### 3.4. Analysis of the Consistency of the Algorithm Used to Select Key Target Genes

The work in [13] demonstrates the fundamental possibility of existence of the key genes in miRNA-target networks. Hereinafter we will refer to them as “true” target genes. Our network-based approach implies that “true” key genes are indexed in the used databases of miRNA targets, and their molecular interactions are known; thus, constructed miRNA-target networks do contain these “true” key genes. In this case the exclusion of random nodes from the network should not greatly change the composition of the key genes. To investigate this, we constructed miRNA-target networks one-by-one for all miRNAs from miRTarBase database, and selected for further analysis 79 miRNAs, which target genes form LCCs with high cardinality (more than 50). For each of these miRNAs, we identified key genes using the described algorithm and referred to them as “true” key genes. Then we randomly excluded 10% of non-key genes from the miRNA-target network and observed that the sets of key genes had changed; we now referred to these changed sets as “test” key genes. For each of the 79 miRNAs, we proceeded with the ejection of non-key genes in nine steps (10% of genes per step) until 90% of genes in each miRNA-target network were excluded. Thus, we obtained nine sets of “test” key genes depending on the size of the network for each of 79 tested miRNAs (Figure 4; the code for this analysis is available on https://github.com/GJOsmak/miRNET_HCM/Code/Node_test.ipynb, accessed on 13 December 2021).

As can be seen from Figure 4, violin plots indicate the positive correlation of the number of “test” key genes with the size of the analyzed network. Moreover, regardless of the network size, more than 95% of the observed “test” key genes belong to the set of the “true” key genes. Thus, we have demonstrated the convergence of our algorithm with the “true” key target genes, assuming that they were indexed in databases we used, and this convergence is independent of a non-key-genes set composition.

## 4. Discussion

In this work, we predicted signaling pathways, regulated by a group of human miRNAs dysfunctional in hypertrophic myocardium, namely hsa-miR-29a, hsa-miR-93, hsa-miR-133a, hsa-miR-199-3p, hsa-miR-221, hsa-miR-222, hsa-miR-451, hsa-miR-497, in order to uncover the pathogenic mechanisms of HCM, proceeding from the oligogenic nature of the disease. It should be noted that the genes encoding these miRNAs were not located in/near known genes with high frequency of occurrence of HCM-associated mutations (such as *MYBPC3*, *MYH7*, *TNNT2*, *TNNI3*, *MYH6*, *MYL2*, *MYL3*, *ACTC1*, *TNNC1*, *TPM1*, *ACTN2*, *CSRP3,* and *MYOZ2*). Thus, differential expression of the selected miRNAs in hypertrophic myocardium can hardly be a reflection of the common regulatory processes shared with these genes.

By analyzing the miRNA-mediated interaction networks, we identified two signaling pathways, enriched by key targets of the studied miRNAs. The first module of three TGFβ-mediated SMAD signaling pathways is regulated by the largest number of selected miRNAs, associated with cardiac hypertrophy: hsa-miR-19b-3p, hsa-miR-21-5p, hsa-miR-29a-3p, hsa-miR-93-5p, hsa-miR-155-5p, hsa-miR-221-3p, hsa-miR-222-3p, and hsa-miR-451a. The second module contains 12 other pathways, each of which is regulated by other miRNAs, namely hsa-miR-1-3p, hsa-miR-133a-3p, hsa-miR-199a-3p, and hsa-miR-497-5p. However, the probability of obtaining such a module by chance was high (*p*-value = 0.11), as opposed to the first module, for which this estimation was at an acceptably low level (*p*-value = 0.0062).

Analysis of our algorithm confirmed its consistency, attesting to the fact that the key genes we describe as involved in TGFβ-mediated SMAD signaling are the “true” key genes with the probability more than 95%. Together with the fact that we were able to identify the TGFβ-mediated SMAD signaling module in the preliminary run of our workflow, when we analyzed a reduced list of miRNAs without miRNA-3p/-5p strand specification (namely, hsa-miR-29a, hsa-miR-93, hsa-miR-133a, hsa-miR-199-3p, hsa-miR-221, hsa-miR-222, hsa-miR-451, and hsa-miR-497), it pointed to the robustness of the chosen network-based approach.

The direct or indirect influence on TGFβ-mediated SMAD signaling under various pathological conditions (although not in HCM) has been described for all miRNAs present in this module with the exception of hsa-miR-222-3p, namely, hsa-miR-19b-3p [22], hsa-miR-21-5p [23], hsa-miR-29a-3p [24], hsa-miR-93-5p [25], hsa-miR-155-5p [26], hsa-miR-221 [27], and hsa-miR-451 [28].

SMAD-dependent signaling is one of main mechanisms to transduce signals of TGF-β cytokine to the cell nucleus, where SMAD2/SMAD3:SMAD4 heterotrimer complex plays a considerable role in regulating the expression of TGF-β-dependent genes [29]. Increase of TGF-β level promotes the pathological remodeling of different tissues [30,31,32]. In human and mice myocardium TGF-β level correlates with fibrosis [33,34,35]. Study of HCM showed that hypertrophy induction in cardiac myocytes promotes collagen expression in fibroblasts via stimulation of TGF-β secretion, therefore contributing to fibrosis [36]. Interestingly, the transcriptional factor Myc, encoded by *MYC* gene, which is the key target of hsa-miR-21-5p, hsa-miR-29a-3p, hsa-miR-93-5p, hsa-miR-155-5p, hsa-miR-222-3p, and hsa-miR-451a, is able to bind *SMAD2* and *SMAD3* and inhibits TGF-β-mediated response [37]. In addition, Myc plays an important role in initiation and maintenance of cardiac hypertrophy, as well as in the processes of apoptosis and angiogenesis (reviewed in [38]). Notably, among selected miRNAs hsa-miR-155-5p interacts with the greatest number of shared key genes we identified that belong to the SMAD-dependent pathway. This well-known proinflammatory miRNA was previously shown to induce cardiac hypertrophy through activation of the AKT and NF-kB signaling pathway [39]. In total, miRNAs via regulation of SMAD-dependent signaling may affect both hypertrophic and fibrotic changes in the heart muscle in HCM.

Overall, using a developed workflow we pinpointed SMAD-dependent and VEGF signaling pathways that contribute in cardiac remodeling and angiogenesis in HCM. Taking into account that both studied miRNAs and their key target genes involved in these pathways are not linked to causative HCM-associated genes, we can assume that their deregulation in HCM most likely occurs in response to the pathogenic processes in myocardium.

## 5. Limitations

One of the limitations of our study is the restricted number of disease-associated miRNAs selected for the analysis. We did not carry out a large-scale search of miRNAs contributing to human cardiac hypertrophy, since the present study represents an interim step between the functional analysis of a single miRNA (proposed earlier) to the holistic analysis of the functional role of all miRNAs deregulated in disease. Our network-based approach is based on the assumption that “true” key genes are indexed in the used databases of miRNA targets, and their molecular interactions are known. However, we cannot exclude the possibility that some of the selected miRNAs are studied insufficiently, and therefore the list of their “true” key genes could be distorted. In total, the success of the proposed approach is highly dependent on the completeness of data on target genes of disease-associated miRNAs. Obviously, increased initial information for subsequent analysis (the number of miRNAs and their targets, data on intermolecular interactions, etc.) will lead to significant improvement of its accuracy. We should also highlight that we considered only pathway modules overrepresented by key target genes of the highest number of selected miRNAs. Taking into account other Reactome pathways connected to the lesser number of miRNAs may also provide valuable data. The results we obtained need to be further investigated using experimental models of hypertrophic cardiomyopathy.

## Figures and Tables

**Figure 1 genes-12-02016-f001:**
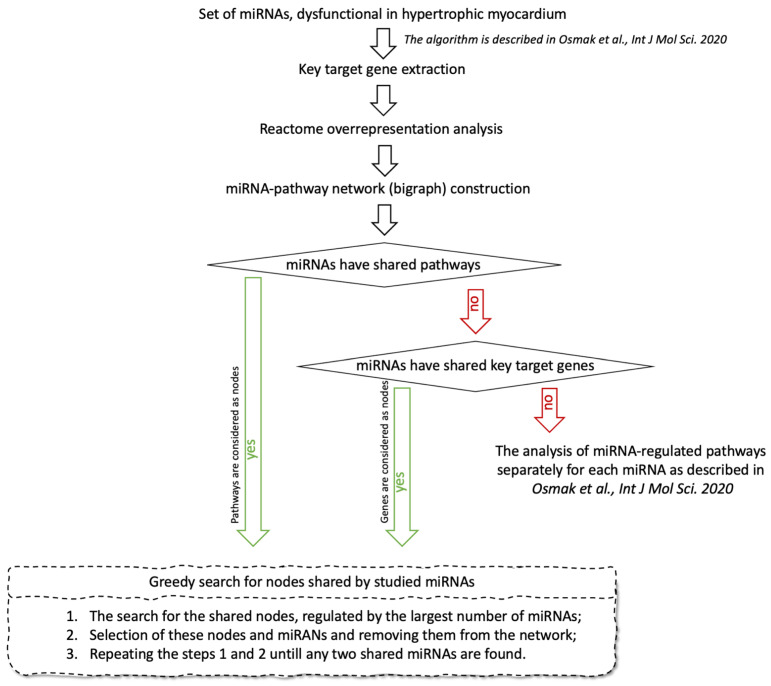
A schematic workflow for the identification of miRNA-regulated pathways involved in HCM pathogenesis.

**Figure 2 genes-12-02016-f002:**
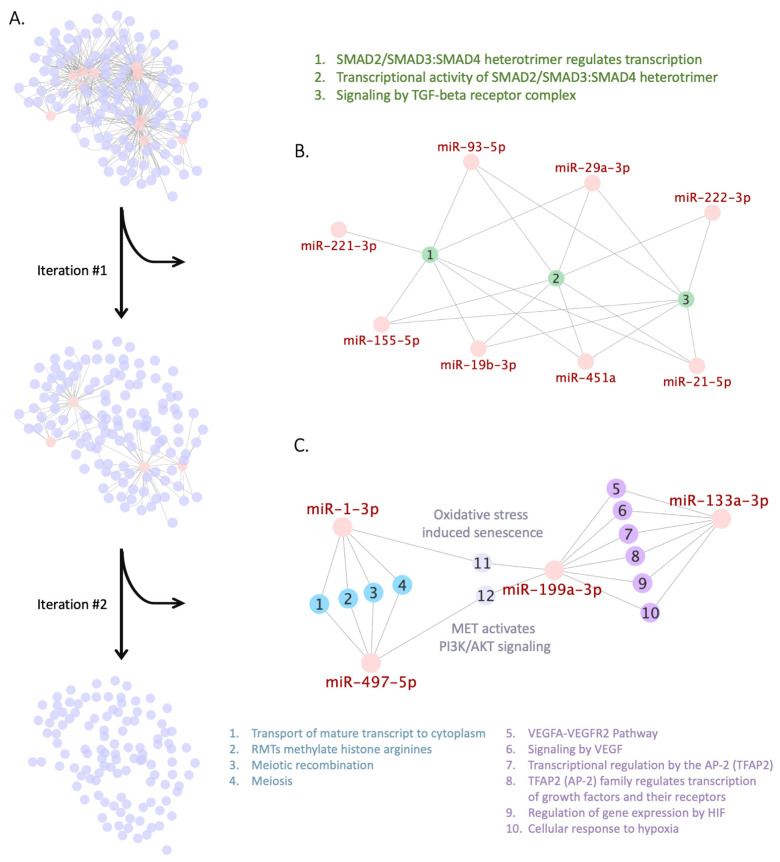
Greedy search for Reactome pathways shared by studied miRNAs. Each network is a bigraph, containing two sets of nodes: a set of miRNAs (henceforward marked in red) and a set of Reactome pathways. Nodes from the miRNA set and the Reactome pathways set are connected by edge if key target genes of a miRNA are overrepresented in a Reactome pathway. (**A**) Two iterations of the greedy search for Reactome pathways connected to the selected miRNAs in the initial miRNA-pathway network. (**B**) The module pinpointed after the first iteration of the greedy search that contains three pathways (nodes and legend marked in green) and eight miRNAs. (**C**) The module pinpointed after the second iteration, which consists of two clusters: first—pathways 1–4 linked to hsa-miR-1-3p and hsa-miR-497-5p (nodes and legend marked in blue) and second—pathways 5–10 linked to hsa-miR-133a-3p and hsa-miR-199a-3p (nodes and legend marked in purple); two “bottleneck” pathways (11 and 12) are linked via hsa-miR-199a-3p to the first cluster and via hsa-miR-497-5p and hsa-miR-1-3p to the second cluster (nodes and legend marked in grey).

**Figure 3 genes-12-02016-f003:**
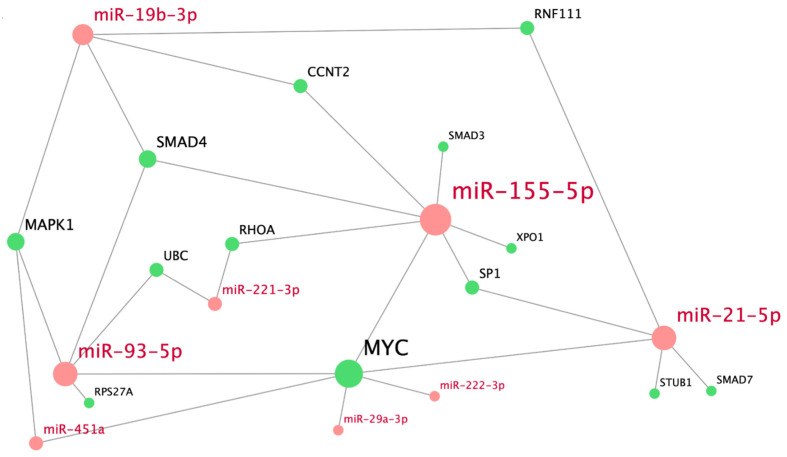
Bigraph of human miRNAs and their key target genes, overrepresented in the module of TGFβ-mediated SMAD signaling pathways. MiRNA nodes are marked in red; nodes of key target genes, in green.

**Figure 4 genes-12-02016-f004:**
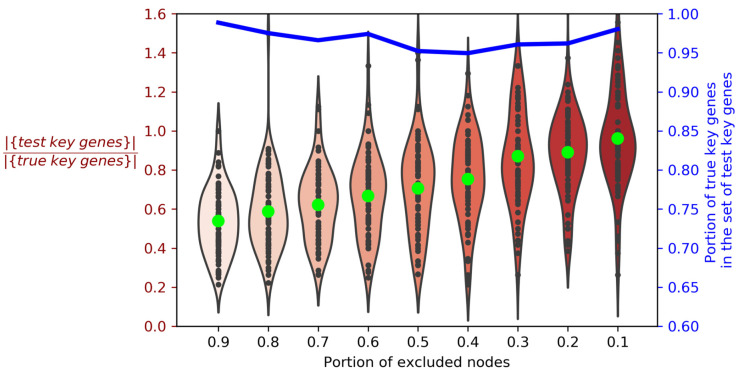
Violin plots illustrating the kernel density distribution for the ratio of miRNAs’ “test” key genes to “true” key genes. The blue line indicates the portion of “true” key genes in the set of “test” key genes. Black dots indicate the ratio of cardinality of the “test” key gene sets to the cardinality of the “true” key gene sets for each analyzed miRNA. Lime points mark their medians. The color gradient is given to facilitate the perception of an increase in the number of “test” key genes. |{set}| is the number of elements in the set.

**Table 1 genes-12-02016-t001:** Number of target genes of analyzed miRNAs, identified in network analysis.

miRNA	Number of Targets			
	Total, *N*	Expressed in Heart Tissue, *N* (% of Total)	Included in LCC *, *N* (% of Total)	Identified as Key Genes of LCC *, *N* (% of Total)
hsa-miR-1-3p	921	504 (55)	181 (20)	70 (8)
hsa-miR-19b-3p	714	425 (60)	108 (15)	13 (2)
hsa-miR-21-5p	612	353 (58)	99 (16)	26 (4)
hsa-miR-29a-3p	265	138 (52)	28 (11)	7 (3)
hsa-miR-93-5p	1220	718 (59)	284 (23)	56 (5)
hsa-miR-133a-3p	131	65 (50)	7 (5)	5 (4)
hsa-miR-155-5p	904	530 (59)	214 (24)	29 (3)
hsa-miR-199a-3p	114	66 (58)	11 (10)	6 (5)
hsa-miR-221-3p	368	207 (56)	57 (15)	10 (3)
hsa-miR-222-3p	394	242 (61)	42 (11)	10 (3)
hsa-miR-451a	31	16 (52)	6 (19)	5 (16)
hsa-miR-497-5p	461	254 (55)	78 (17)	13 (3)

* LCC: largest connected component.

## Data Availability

The code is available at GitHub: https://github.com/GJOsmak/miRNET_HCM, accessed on 13 December 2021.

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
