# Peer review of "MiRNA-Regulated Pathways for Hypertrophic Cardiomyopathy: Network-Based Approach to Insight into Pathogenesis"

_genes, 2021, doi:10.3390/genes12122016_

Round 1

Reviewer 1 Report

Osmack et al describe a bioinformatic strategy for identifying pathways potentially targeted by miRNAs involved in cardiac hypertrophy. My concern is that while this information is useful as a starting point, it is based on other large scale databases and studies (Osmack 2020 and the Human Protein Atlas). In Osmack 2020 only a small subset of identified factors were found to actually be expressed in the mouse heart. And while the Human protein Atlas is based on immunohistochemistry experiments, their mRNA data is from large scale sequencing. What is lacking is at least some form of validation for perhaps a few of the pathway targets identified. Do they actually physically interact with the miRNAs described? Do these miRNAs actually regulate the pathway targets?

Otherwise, I think it is a great tool for the scientific community which could be utilized with greater confidence with validation.

Author Response

Dear Reviewer, thank you for the appreciation of our manuscript and for raising such important issues. Please, find below the responses to your comments. All changes, made to the manuscript according to your comments, are highlighted in yellow.

To select target genes of analyzed miRNAs we used the MiRTarBase database (http://mirtarbase.cuhk.edu.cn/) which accumulates only data on experimentally validated miRNA-target interactions. In more details, all these genes were shown to physically interact with studied miRNAs by different methods such as reporter assay, western blot, microarray and/or next-generation sequencing experiments. We now reflected this clarification in the “Tools and Databases”:

“The MiRTarBase database (http://mirtarbase.cuhk.edu.cn/) was used to select experimentally validated direct target genes of miRNAs, described for humans”.

Available literature data support the evidence on the involvement of the studied miRNAs in regulation of SMAD- and VEGF-dependent pathways in the pathological conditions. We have now described it in the “Discussion” section:  

The direct or indirect influence on the detected signaling pathways under various pathological conditions (although not in HCM) has been described for these miRNAs except for hsa-miR-222, namely, for hsa-miR-29 [30], hsa-miR-93 [31], hsa-miR-451 [32], hsa-miR-497 [33], hsa-miR-221 [34], hsa-miR-199a-3p [35], and hsa-miR-133a [36].

The presented data most likely point to the validity of the results obtained. However we cannot but agree that experimental validation of the pathway targets is definitely needed to strengthen the developed network-based approach that is what we wrote about in the “Limitations” section in the submitted manuscript. Since we are interested in hypertrophic cardiomyopathy we are currently working on choosing its proper model to get started with our experimental work and expect to report the results as soon as possible.

Reviewer 2 Report

This is an interesting analysis that utilises current miRNA data into pathway discovery for HCM. 

Comments:

  • Please describe the process that you have followed to select which miRNAs to use in this study (e.g. systematic literature search, keywords, date of search etc) in the methods. Also describe inclusion, exclusion criteria used for studies. 
  • Are you able to provide a 95% confidence interval in the comparison of the expected number of signaling pathways vs what was actually found (with bootstrapping)?
  • Have you considered taking into account the ethology of HCM in the cohorts that the miRNAs were identified? Is it possible that certain miRNAs are specific to the affected protein (considering a pathogenic or likely pathogenic variant)?
  • Does heavy involvement of a pathway from your analysis suggest a causative correlation or could it just be a response mechanism by the cardiomyocytes? Are there ways to study this?

Author Response

Dear Reviewer, thank you for the careful and thoughtful analysis of our manuscript and for the remarks of essential significance. We are sincerely grateful for all your advice. Please, find below the responses to each of your comments, step by step.  All changes in the revised manuscript made in accordance with your recommendations are marked in grey.

Please describe the process that you have followed to select which miRNAs to use in this study (e.g. systematic literature search, keywords, date of search etc) in the methods. Also describe inclusion, exclusion criteria used for studies. 

Thank you for pointing out this defect. As we described in “Limitations” section we did not carry out a comprehensive search of miRNAs contributing to human cardiac hypertrophy, and the present study represents an interim step between the functional analysis of a single miRNA (proposed earlier) to the large-scale analysis of the functional role of all miRNAs deregulated in disease. However, you are definitely right that the procedure of selecting the miRNAs for our analysis should be clarified in the text. Therefore, we now added the following paragraph in the “Results” section:

“In order to select miRNAs for the analysis we performed a search of studies, which were indexed by terms “cardiac hypertrophy”, “myocardium”, “miRNA expression”, in the PubMed database published by June 2021. Only miRNAs investigated in the myocardial tissue in two or more studies were considered. Among them miRNAs hsa-miR-29a, hsa-miR-93, hsa-miR-133a, hsa-miR-199-3p, hsa-miR-221, hsa-miR-222, hsa-miR-451, and hsa-miR-497 shown to be differentially expressed in at least one of the studies, were selected for the subsequent analysis.”

Are you able to provide a 95% confidence interval in the comparison of the expected number of signaling pathways vs what was actually found (with bootstrapping)?

Thank you for this remark. We now provided a 95% confidence interval for the performed analysis and reflected it in the second table of Supplementary. This estimate was also obtained using bootstrap.

Have you considered taking into account the ethology of HCM in the cohorts that the miRNAs were identified? Is it possible that certain miRNAs are specific to the affected protein (considering a pathogenic or likely pathogenic variant)?

Thank you for this question. Unfortunately, there are currently too few publications on the involvement of miRNAs in HCM that take into account the genetic characteristics of patients recruited; therefore the etiology of HCM is mostly out of focus. We checked the genomic location of selected miRNAs’ genes. All they were not located in/near known genes with high frequency of occurrence of HCM-associated mutations (such as MYBPC3, MYH7, TNNT2, TNNI3, MYH6, MYL2, MYL3, ACTC1, TNNC1, TPM1, ACTN2, CSRP3 and MYOZ2). Moreover, the key target genes of these miRNAs, pinpointed with developed network-based approach, were also not among abovementioned HCM-associated genes. Thus, we believe that selected miRNAs are not specific to genes with pathogenic or likely pathogenic variants and may be mostly involved in HCM independently from causative mutations. We now specify this in text:

It should be noted that the genes encoding these miRNAs were not located in/near known genes with high frequency of occurrence of HCM-associated mutations (such as MYBPC3, MYH7, TNNT2, TNNI3, MYH6, MYL2, MYL3, ACTC1, TNNC1, TPM1, ACTN2, CSRP3 and MYOZ2). Thus differential expression of the selected miRNAs in hypertrophic myocardium can hardly be a reflection of the common regulatory processes shared with these genes.

We also reflected this aspect in the last paragraph of “Discussion” (see the answer to the next question).

Does heavy involvement of a pathway from your analysis suggest a causative correlation or could it just be a response mechanism by the cardiomyocytes? Are there ways to study this?

This is a very important point. Taking into account this and previous questions we modified the ending of the “Discussion” section with the following sentence:

Taking into account that both studied miRNAs and their key target genes involved in these pathways are not linked to causative HCM-associated genes, we can assume that their deregulation in HCM most likely occurs in response to the pathogenic processes in myocardium.”